# Ferritin-Based Single-Electron Devices

**DOI:** 10.3390/biom12050705

**Published:** 2022-05-15

**Authors:** Jacqueline A. Labra-Muñoz, Arie de Reuver, Friso Koeleman, Martina Huber, Herre S. J. van der Zant

**Affiliations:** 1Department of Physics, Huygens-Kamerlingh Onnes Laboratory, Leiden University, Niels Bohrweg 2, 2300 RA Leiden, The Netherlands; huber@physics.leidenuniv.nl; 2Kavli Institute of Nanoscience, Delft University of Technology, Orentzweg 1, 2628 CJ Delft, The Netherlands; a.r.dereuver@gmail.com (A.d.R.); F.m.koeleman@student.tudelft.nl (F.K.)

**Keywords:** ferritin, Coulomb blockade, single-electron transport, nanoelectronics

## Abstract

We report on the fabrication of single-electron devices based on horse-spleen ferritin particles. At low temperatures the current vs. voltage characteristics are stable, enabling the acquisition of reproducible data that establishes the Coulomb blockade as the main transport mechanism through them. Excellent agreement between the experimental data and the Coulomb blockade theory is demonstrated. Single-electron charge transport in ferritin, thus, establishes a route for further characterization of their, e.g., magnetic, properties down to the single-particle level, with prospects for electronic and medical applications.

## 1. Introduction

Ferritin, the iron storage protein, has drawn the attention of the scientific community in recent decades. Its unique structure consists of a spherical shell composed of 24 peptide subunits [1]. In mammals, two types of subunits are present: a heavy (21 kDa) and a light (19 kDa) chain [2]. The heavy chain is responsible for the oxidation of Fe(II) to Fe(III). The light chain fosters nucleation and the storage of Fe(III) as a mineral mixture consisting mainly of ferrihydrite, storing iron in a nontoxic form [1]. Dysfunctional ferritin has been related to neurodegenerative diseases for different reasons. First, it allows unwanted egress of iron from the ferritin, which increases the available amount of free iron that is toxic to cells, causing oxidative stress injury [1,3]. Second, the ferritin core’s composition in the brain of patients affected by these diseases has been found to be altered [4,5]. The mechanism leading to dysfunctional ferritin remains unclear; therefore, a better understanding of the physical properties of individual ferritin particles is vital.

Published studies on the electrical characterization of ferritin have mostly been performed on ferritin networks and monolayers [6,7,8,9,10,11,12,13,14,15,16,17,18,19,20,21,22,23]. Almost 20 years ago, Xu et al. [7] reported that holoferritin networks, i.e., ferritin with iron core, are 5–15 times more conductive than the networks of the iron-depleted version, apoferritin [7]. Although Xu et al. reported S-shaped current vs. voltage (*IV*) characteristics, other phenomena have been observed, such as switching [10] and negative differential resistance [17]. The charge transport through ferritin monolayers has been studied by Kumar et al. [6], who reported different transport mechanisms including direct tunneling, sequential tunneling, and hopping, depending on the ferritin iron content. Ferritin multilayers have been studied as well [20], finding highly correlated electron transport, consistent with electron transport in a quantum dot. Bera et al. [14] reported similar *IV*s for six layers of holo- and apoferritin. Targeted applications range from multilayered gate dielectrics [8], to metal-insulator nanocomposites [13], high-efficiency solar energy conversion [11], bio-nanobatteries [12], to p-n junctions [15], among others. From a medical perspective, Holovchenko et al. [21] showed that the average conductance of Alzheimer’s ferritin networks is about two orders of magnitude lower than that of control ferritin, suggesting the use of ferritin resistance measurements as a diagnostic tool.

Studies focused on the electrical characterization of single ferritin are scarce. These studies are based on either scanning tunneling microscopy (STM) [18,22,24] or atomic force microscopy (AFM) [7,25,26] and are performed at room- or higher temperatures. Single holoferritin was found to be more conductive than apoferritin by AFM [25], as found in networks. In contrast, Rakshit et al. [24] obtained similar *IV* characteristics for apo- and holoferritin by STM. In this work, single horse-spleen ferritin particles are studied by trapping them in self-aligned nanogaps and measuring their electronic properties at low temperatures. We find that the devices are unstable at room temperature, but below 100 K they allow for the acquisition of reproducible data that establishes the Coulomb blockade as the main transport mechanism through them.

## 2. Results

We used commercial horse-spleen ferritin purchased from Sigma Aldrich. Figure 1a shows a schematic representation of ferritin. It consists of a mineral core surrounded by an organic shell with an outer diameter of 12 nm, and a shell thickness of 2 nm, approximately. From the same batch used in the conductance measurements, we determined the size distribution of ferritin cores through transmission electron microscopy (TEM); see Appendix A for details. The analysis of 1502 particles showed that the cores’ size varied from 4 to 8.6 nm, as shown in Appendix A.

Self-aligned nanogaps were fabricated as reported in the Methods section, following published fabrication routes [27,28]. Figure 1d shows a scanning electron microscopy (SEM) image of a self-aligned nanogap before ferritin deposition. The distance between the source and drain electrodes (gap length) varied between 8 and 25 nm, while the gap width was 10 μm.

Prior to ferritin deposition, we recorded at room temperature and in a vacuum ($10^{-4}$ mbar) the current vs. voltage (*IV*) characteristics of each electrode pair; some samples were also checked at helium temperature. Devices were disregarded that exhibited a current greater than the noise floor (2 pA) over the bias voltage range probed (±1 V and ±400 mV, for larger (11–25 nm) and smaller gaps (8–13 nm), respectively). Thus, we only selected open gaps for further studies with ferritin particles (see Figure 1c, grey curve).

Deposition was performed by drop-casting 2–4 μL of ferritin solution diluted 200 times (∼270 μg/mL) onto the electrodes, followed by immediate vacuum pumping. Figure 1b shows a schematic representation of a gap after ferritin deposition. We identified the presence of ferritin trapped within the gap by comparing the *IV* characteristics of the gap before and after deposition, measured in a vacuum. Figure 1c shows representative examples of *IV* curves, measured before (gray curve) and after (green curve) deposition at room temperature for device A1. After deposition, a clear increase in current was seen, but the *IV* was unstable with switches between higher and lower conductive states. Furthermore, hysteretic behavior was also often observed while sweeping the bias voltage up and down. This behavior was generally observed for other samples as well.

The ferritin-deposited devices were cooled down to temperatures close to T=4.2 K. At these temperatures, the *IV* characteristics were stable, and the hysteresis was no longer present. Appendix A summarizes the four behaviors detected after ferritin deposition, at temperatures close to 4.2 K. The behaviors were: highly conductive linear *IV*s (most likely ferritin aggregates); still open gaps (no ferritin trapped); tunneling; and Coulomb-blockade (CB)-like *IV*s. Appendix A shows the reference measurements, using only the buffer solution; after three buffer depositions, no CB-like features were detected.

Here, we focused on analyzing the CB-like *IV* characteristics, since they may represent devices with one particle trapped (see below). Figure 2 shows two typical CB-like *IV*s recorded at 4.2 or 5 K, displaying clear step-like features (light blue dots) or a single transport gap centered around zero bias (green dots). For clarity and to facilitate the simulations, these *IV* curves were either the descendant curves of the *IV* cycles, i.e., the current recorded from 50 mV to −50 mV, or the ascendant ones. In some cases, abrupt *IV* changes, especially in conductivity, were observed after thermal, voltaic, and temporal cycling (for details, see Appendix A). An example of this effect is shown in Figure 2; device D1 (Figure 2g) was recorded first, at 4.2 K. The same device was measured at 5 K again (Figure 2a), after warming it up to 250 K. The current increased by three orders of magnitude, and the IV curve showed clear steps.

We used the orthodox Coulomb Blockade (CB) model [30] to describe the *IV* characteristics, measured at the lowest recorded temperature as shown in Figure 2. The model considered single-electron tunneling, without incorporating second-order processes such as tunneling through virtual states and cotunneling. A Matlab script was written with $R_{1}$, $R_{2}$, $C_{1}$, $C_{2}$, $Q_{0}$, and *T* as the simulation parameters; the parameters are defined in the caption of Table 1. The temperature was taken to be the measured temperature near the sample and was not adjusted in the simulations. The black dashed lines shown in Figure 2 are the CB fits to the data; the simulation parameters are listed in Table 1. The consistency between the data and simulations is visible both in the *IV* and the differential conductance (d*I*/d*V*) plots. The excellent agreement indicates that the dominant transport channel is indeed through a single particle, although three-terminal measurements with a gate are needed to give a conclusive answer.

To further investigate the validity of the CB model, we measured *IV*s at different temperatures. Figure 3 shows *IV*s obtained at different temperatures, on device A1. At 4.2 K (blue light dots), the *IV*s present clear Coulomb-blockade steps. At 22 K (orange dots), steps are no longer visible, and the blockade part of the *IV*s is linear, i.e., the blockade has been lifted. The black dashed lines are the Coulomb-blockade fits to the data, using the same simulations parameters $R_{1}$, $R_{2}$, $C_{1}$, $C_{2}$, and $Q_{0}$, while adjusting only the temperature.

In total, 22 devices displayed Coulomb-blockade like IVs. Appendix A shows the experimental IVs and the CB fits of all data. The simulation parameters are collected in Appendix A. Figure 4 summarizes the total capacitances and resistances used for the simulations. A broad dispersion was found in the parameters: the total capacitance varied over two orders of magnitude from 0.84 to 62.7 aF, while the total resistance varied over four orders of magnitude from 0.02 GΩ to 72 GΩ.

## 3. Discussion

To place the values of the fit parameters in a broader context, we compared them with values found in other experiments. For ferritin, a resistance of 2 GΩ was reported [24] by STM, but, a much higher resistance was reported by AFM (25 GΩ–1.8 TΩ) [7,26,31]. Our experimental resistances were closer to that reported by STM and to the lower range obtained by AFM. Capacitance values have not been reported for single ferritin particles in the literature.

We note that the measured core distribution can lead to a sizable dispersion in the capacitances/resistances but cannot fully account for the observed dispersion in Appendix A and Figure 4. However, different geometrical arrangements are discussed below, which reproduce this broad dispersion. Figure 4c indicates that high capacitances were found for low resistances. This trend is consistent with geometrical considerations in which good electrical contacts (low resistances) imply large electrode-particle capacitances. More specifically, the particle could lie symmetrically in between the electrodes (case I; see Appendix A) or on top of them (case II). If the gap between the electrodes is larger than the ferritin size, the particle could couple asymmetrically to the electrode if one side of the particle makes direct contact with one of the electrodes, while there is some space in between the particle and the other electrode. This results in two additional cases to be considered: asymmetric contact with the particle laying completely between the electrodes (case III) or on top of them (case IV).

For these four different cases we calculated the capacitances using a parallel plate model, by considering $\epsilon_{r}$ to be between 10 and 20 [32,33], see Appendix A for details. The estimates are shown in Appendix A, and the minimum estimate for the total capacitance (0.4 aF) was in agreement with the minimum experimental capacitance (0.84 aF). The maximum estimated capacitance (11.3 aF) was about five times lower than the maximum experimental capacitance (62.7 aF), although only two capacitances were found to be larger than 24 aF. Ways to account for this difference could be to consider a barrier thickness thinner than 2 nm or a higher dielectric constant.

Within the CB model, the total resistance (*R*) of an individual ferritin particle in a gap between source and drain contacts, comprised two tunnel resistances in series, from the ferritin core to each electrode, i.e., $R=R_{1}+R_{2}$. Each of these resistances can be expressed as $R_{1,2}=R_{0}\exp\left(\beta d\right)$, where $R_{0}$ is an effective contact resistance, *d* is the thickness of the dielectric, and β is the exponential distance decay factor. For β we take 0.26 A${}^{-1}$, which is the average reported value for proteins [34].

The calculated resistances in this model are again listed in the SI. We found that the maximum experimental resistance (72 GΩ) was within the range of our estimates (0.17–126.55 GΩ). The minimum experimental resistance (0.02 GΩ) was somewhat lower than our minimum estimate. A possible reason is that β is smaller than what we assumed for ferritin. For example, if we consider β=0.21 Å${}^{-1}$, the minimum resistance (case I) is 0.06 GΩ, which is on the same order of magnitude as the smaller experimental resistance.

In conclusion, we demonstrated that individual ferritin particles can be trapped in self-aligned nanogaps. The observed instability at room temperature was suppressed at low temperature, and clear Coulomb-blockade effects were observed including Coulomb staircases. Excellent fits of the current–voltage characteristics to the Coulomb blockade model strongly indicate that transport is through an individual particle. Estimates based on different arrangements of the particle in between the electrodes and reported values for transport through proteins reproduced the capacitance and resistance values determined from the CB model. Thus, the experiments showed that ferritin particles act as a model CB system, and in a next step, it would be interesting to relate the transport characteristics to specific ferritin properties. This, for example, includes a better understanding of the unstable behavior at elevated temperatures and of the magnetic properties of the core. The latter can be studied using a three-terminal device with a gate in combination with magnetic field measurements.

## 4. Methods

### 4.1. Ferritin Purity Assessment

Commercial horse-spleen ferritin purchased from Sigma Aldrich (Cat. No. 270-40, Lot: 08E1805) of 54 mg/mL protein concentration was used with no further purification. For the conductance measurements, the ferritin solution was diluted by a factor of 200 using a buffer solution (0.15 M NaCl and 10 mM Tris, pH 8.0, 0.1% sodium azide). The final solution had a protein concentration of 270 μg/mL with >95% purity as assessed by sodium dodecyl sulfate–polyacrylamide gel electrophoresis (Appendix A). The homogeneity of the solution was assessed by the dynamic light scattering technique (DLS), which is reported in Appendix A (Appendix A).

### 4.2. Ferritin Size Statistics

The ferritin mineral cores were examined without further staining in a JEM-1400 transmission electron microscope (TEM). To further inspect the protein shell structure, uranyl acetate (2% solution) was used. For more details, refer to Appendix A (Appendix A). A home-made Matlab script was used for particle detection and size statistics.

### 4.3. Device Fabrication

The devices were fabricated as follows. On top of a doped Si/SiO${}_{2}$ substrate, the first electrode was defined by e-beam lithography (EBL) and evaporation of 5 nm of titanium (adhesive layer) and subsequently 25 nm of platinum. On top of the platinum layer, a 20 nm chromium layer was deposited. Upon oxidation chromium expands its size. In this manner, chromium oxide acted as a shadow mask of a few nanometers near the edge of the first electrode. The thickness of the chromium layer determined the expansion and, thus, the size of the gap. A second EBL cycle defined the second electrodes, by depositing 5 nm of titanium and 20 nm of platinum. In the final step, the chromium layer was etched away (wet-etch step) to reveal the underlying nanogaps.

### 4.4. Electrical Measurements

The electrical measurements of most of the devices were performed in a variable temperature insert (VTI) with TU Delft home-built low-noise electronics. The noise was about 1–2 pA at room temperature and below 1 pA at 4.2 K. A 1-K pot allowed measurements below 4.2 K for devices J1, J2, and K1. Device H1 was measured in a dilution refrigerator equipped with the same TUDelft home-built electronics and a similar noise level.

### 4.5. Coulomb-Blockade Fits

The *IV*s obtained showing CB-like features were fitted to the CB orthodox theory, using a home-made Matlab script that is available upon request. The optimal fit was determined by the user, i.e., by visual inspection.

## Figures and Tables

**Figure 1 biomolecules-12-00705-f001:**
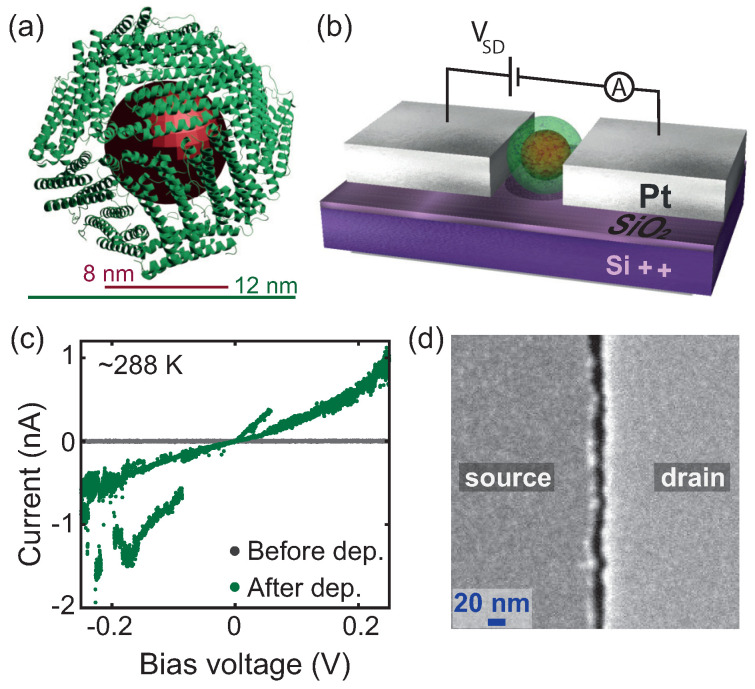
(**a**) Schematic representation of ferritin, based on the protein data base (PDB) of horse-spleen apoferritin (PDB ID: 2W0O [29]). The organic shell is in green; the mineral core occupying the internal cavity is in red. (**b**) Schematic circuit of a device containing one ferritin particle. (**c**) Electrical characterization of device A1 before (grey curve) and after (green curve) ferritin deposition, measured at room temperature, in vacuum. The grey curve indicates an open circuit, reflecting an empty device. The increase in current shown in the green curve indicates the capture of ferritin. In both cases, the current is measured by sweeping the voltage from negative to positive values, followed by sweeping from positive to negative values. (**d**) Scanning electron microscopy image of an empty device showing the gap of 9–19 nm size between source and drain electrodes.

**Figure 2 biomolecules-12-00705-f002:**
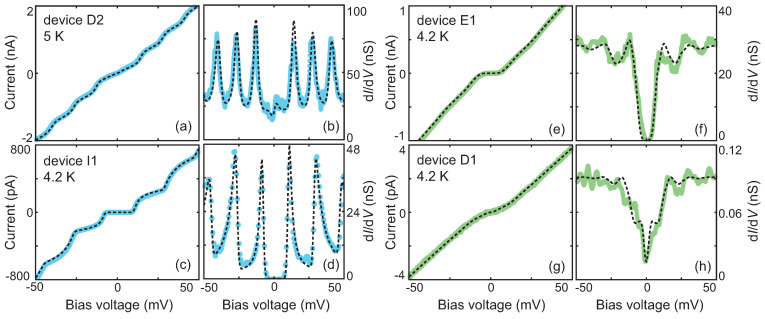
Experimental current-voltage (*IV*) characteristics (colored dots) and the corresponding calculated curves using the orthodox Coulomb blockade model (black dashed lines) acquired on four devices, displaying clear step-like features (light blue dots) or a single transport gap centered around zero bias (green dots). (**a**,**e**,**g**) Experimental *IV*s measured on, respectively, device D2, E1, and D1; obtained by sweeping the voltage from positive to negative values. (**c**) *IV* measured on device I1, obtained by sweeping the voltage from negative to positive values. The corresponding differential conductance is depicted in figures (**b**,**d**,**f**,**h**). The data were recorded at either 4.2 or 5.0 K. The parameters used for the simulations are presented in Table 1.

**Figure 3 biomolecules-12-00705-f003:**
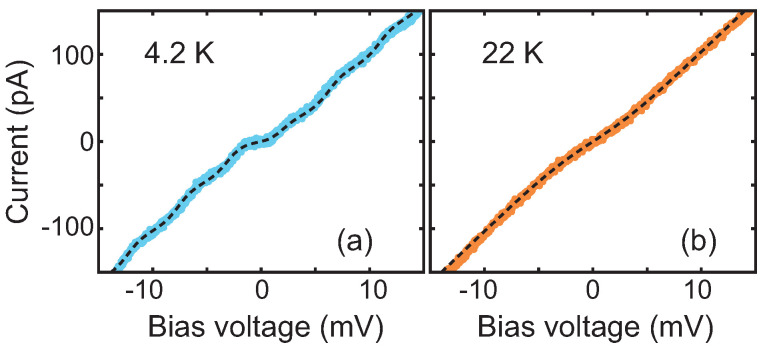
Experimental current-voltage characteristics acquired on device A1, at two different temperatures. (**a**) Data measured at 4.2 K (light blue dots). (**b**) Data acquired at 22 K (orange dots). The black dashed lines indicate the calculated curves using the orthodox Coulomb blockade model with parameters: $C_{1}=28$ aF, $C_{2}=34.7$ aF, $R_{1}=0.4$ MΩ, $R_{2}=0.08$ GΩ, $Q_{0}=0.15$ e.

**Figure 4 biomolecules-12-00705-f004:**
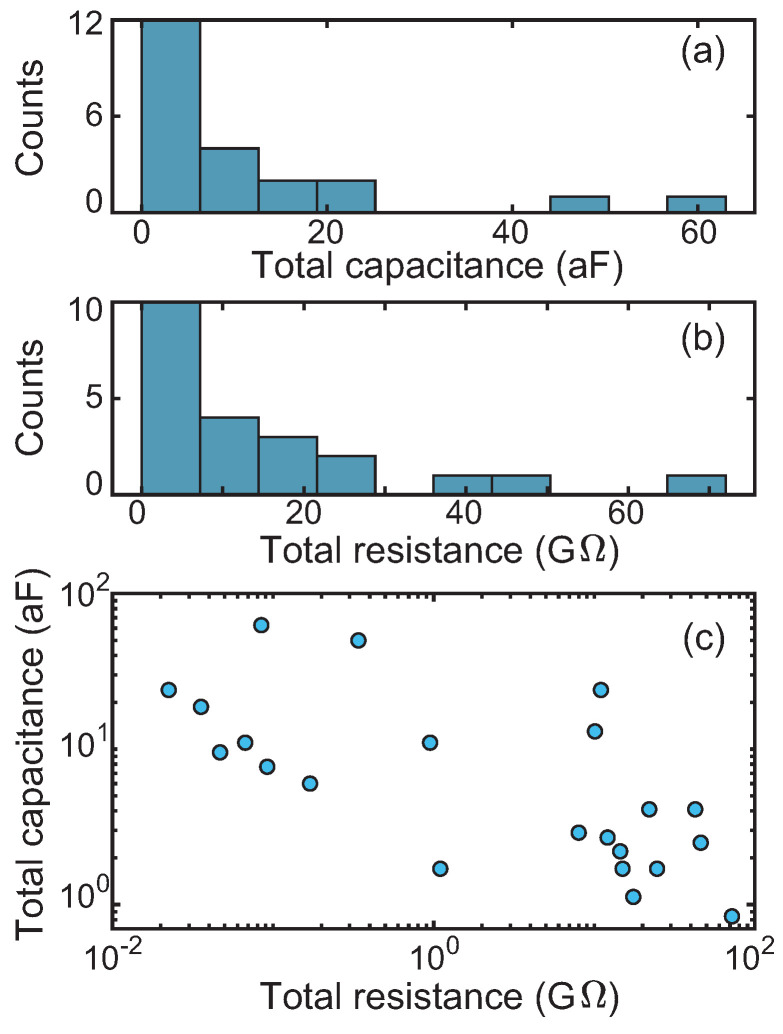
Total capacitances ($C=C_{1}+C_{2}$) and total resistances ($R=R_{1}+R_{2}$) calculated with the orthodox Coulomb blockade model for the 22 devices showing Coulomb-blockade at temperatures below 100 K. Appendix A contains the entire set of Coulomb blockade parameters used to model each device, at the lowest recorded temperature. (**a**) Total capacitance histogram. (**b**) Total resistance histogram. (**c**) Total capacitance plotted against the total resistance.

**Table 1 biomolecules-12-00705-t001:** Coulomb-blockade simulation parameters used to generate the four simulated *IV*s depicted in Figure 2. $C_{1}$ and $C_{2}$ are the junction capacitances on the left and right sides, respectively. $R_{1}$ and $R_{2}$ are the tunnel resistances on the left and right sides, respectively. $Q_{0}$ is the offset charge, and *T* is the temperature.

Device	$C_{1}$ (aF)	$C_{2}$ (aF)	$R_{1}$ (MΩ)	$R_{2}$ (MΩ)	$Q_{0}$ (e)	*T* (K)
D2	12.0	12.0	0.4	22.0	−0.55	5.0
I1	0.8	8.7	8.6	38.2	−0.06	4.2
E1	9.5	9.2	17.8	17.8	−0.12	4.2
D1	12.0	12.0	5500.0	5500.0	−0.35	4.2

## Data Availability

Data supporting the findings of this paper are available in 4TU.ResearchData, see doi:10.4121/19306865.

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
