# Peer review of "Ferritin-Based Single-Electron Devices"

_biomolecules, 2022, doi:10.3390/biom12050705_

Round 1
Reviewer 1 Report
The manuscript describes a fabrication and electrical characterization of devices utilizing horse-spleen ferritin particles as single electron transport channels. Authors demonstrated good agreement between the experimental data and orthodox Coulomb blockade through a single particle in the samples presented. It would be interesting to see a further development of the devices studied in this manuscript using a three-terminal configuration with a gate.
Author Response
Point 1: The manuscript describes a fabrication and electrical characterization of devices utilizing horse-spleen ferritin particles as single electron transport channels. Authors demonstrated good agreement between the experimental data and orthodox Coulomb blockade through a single particle in the samples presented. It would be interesting to see a further development of the devices studied in this manuscript using a three-terminal configuration with a gate.
Response: We thank the referee for his/her positive evaluation and agree that three-terminal measurements would be interesting. In fact, we are presently working on establishing three-terminal ferritin devices with a gate. We have added a sentence in the text to make clearer how the gate could contribute to a better understanding.

Reviewer 2 Report
Labra-Muñoz, van der Zant, and collaborators present a detailed investigation of single-molecule transport properties of holoferritin using self-aligned nanogaps. The results are in agreement with a Coulomb-blockade transport mechanism, with a broad distribution of parameters. The authors show that the observed variations are consistent with different adsorption geometries, which lead to dramatic changes in capacitance and resistance.
The paper is well-written and deserves to be published essentially without changes. The authors might want to add a comment on the expected effect of the core size, which varies from 4 to 8.6 nm in the particular batch used for conductance measurements.
Author Response
Point 1: Labra-Muñoz, van der Zant, and collaborators present a detailed investigation of single-molecule transport properties of holoferritin using self-aligned nanogaps. The results are in agreement with a Coulomb-blockade transport mechanism, with a broad distribution of parameters. The authors show that the observed variations are consistent with different adsorption geometries, which lead to dramatic changes in capacitance and resistance.
The paper is well-written and deserves to be published essentially without changes. The authors might want to add a comment on the expected effect of the core size, which varies from 4 to 8.6 nm in the particular batch used for conductance measurements.
Response: We thank the referee for his/her positive evaluation and we agree that that could be useful and have added the following text in lines 124 to 127 of the manuscript: “We note that the measured core distribution can lead to a sizable dispersion in the capacitances/resistances but cannot fully account for the observed dispersion in Table S1 and Fig. 4. However, different geometrical arrangements are discussed below which reproduce this broad dispersion.“
